# Chemical Analysis of Gunpowder and Gunshot Residues

**DOI:** 10.3390/molecules28145550

**Published:** 2023-07-20

**Authors:** Miguel Serol, Samir Marcos Ahmad, Alexandre Quintas, Carlos Família

**Affiliations:** 1Molecular Pathology and Forensic Biochemistry Laboratory, Centro de Investigação Interdisciplinar Egas Moniz (CiiEM), Instituto Universitário Egas Moniz (IUEM), Campus Universitário—Quinta da Granja, Monte da Caparica, 2829-511 Caparica, Portugal; miguel.serol.99@gmail.com (M.S.); smahmad@egasmoniz.edu.pt (S.M.A.); aquintas@egasmoniz.edu.pt (A.Q.); 2Forensic and Psychological Sciences Laboratory Egas Moniz, Campus Universitário—Quinta da Granja, Monte da Caparica, 2829-511 Caparica, Portugal

**Keywords:** chemometrics, chromatography, gunpowder, gunshot residue, spectroscopy

## Abstract

The identification of firearms is of paramount importance for investigating crimes involving firearms, as it establishes the link between a particular firearm and firearm-related elements found at a crime scene, such as projectiles and cartridge cases. This identification relies on the visual comparison of such elements against reference samples from suspect firearms or those existing in databases. Whenever this approach is not possible, the chemical analysis of the gunpowder and gunshot residue can provide additional information that may assist in establishing a link between samples retrieved at a crime scene and those from a suspect or in the identification of the corresponding model and manufacturer of the ammunition used. The most commonly used method for the chemical analysis of gunshot residue is scanning electron microscopy with energy dispersive X-ray, which focuses on the inorganic elements present in ammunition formulation, particularly heavy metals. However, a change in the legal paradigm is pushing changes in these formulations to remove heavy metals due to their potential for environmental contamination and the health hazards they represent. For this reason, the importance of the analysis of organic compounds is leading to the adoption of a different set of analytical methodologies, mostly based on spectroscopy and chromatography. This manuscript reviews the constitution of primer and gunpowder formulations and the analytical methods currently used for detecting, characterising, and identifying their compounds. In addition, this contribution also explores how the information provided by these methodologies can be used in ammunition identification and how it is driving the development of novel applications within forensic ballistics.

## 1. Introduction

Security is at the forefront of public policy, being one of the primary concerns of law enforcement agencies across the globe. A widely recognised threat to safety is firearms [1,2], as their use or misuse is responsible for many deaths yearly. For example, in 2019, there were 250,000 estimated deaths related to firearms worldwide, of which approximately 177,000 resulted from physical violence, 20,000 from unintentional firearm cases, and 53,000 from self-harm. In the USA alone, in 2019, there were approximately 37,000 deaths associated with firearms, of which 13,000 resulted from physical violence, about 650 from unintentional firearm cases, and 23,000 from self-harm. By comparison, in the European Union (EU), where the purchase of firearms is heavily regulated [3], in the same year, there were almost 6500 deaths associated with firearms, of which around 1000 resulted from physical violence by firearms, 380 from unintentional firearm cases, and 5000 from self-harm by firearms [4]. The sheer number of occurrences worldwide emphasises the need for a reliable framework for the forensic analysis of evidence derived from firearms use.

Ballistics is the discipline dedicated to study of the movement and behaviour of projectiles through the air. This study may focus on different parts of the projectile trajectory, allowing the division of this discipline into internal ballistics, external ballistics and terminal ballistics [5,6,7,8]. The forensic analysis of firearms relies primarily on forensic ballistics, which comprehends the visual examination of engraved or printed marks on firearm-related elements, such as those on projectiles and casings, and subsequent comparison against samples obtained from a suspect firearm or references in databases [5,6,7,8,9]. Whenever this physical analysis is not possible, the chemical analysis of gunshot residue (GSR) may assist in the investigation since it provides additional information that can provide the basis for estimating the firing distance and time since discharge, or even the identification of the corresponding model and manufacturer of the ammunition [5,10,11], which can ultimately lead to the identification of the suspect [11,12,13,14,15,16,17,18,19].

The current contribution reviews the composition of primer and gunpowder used in modern ammunitions and the chemical methodologies currently used for detecting, characterising, and identifying their compounds and elements. The present contribution will explore how the information that these methodologies provide can help in firearm identification and how these data drive the development of novel applications within forensic ballistics, particularly when associated with complex data analysis and statistical modelling. However, sample collection procedures will not be explored. For a detailed review on that topic, please refer to Shrivastava et al. [14] or Goudsmits et al. [15].

## 2. Main Components of Gunpowder and Gunshot Residue

The most common type of ammunition currently in use is single-use cartridges, which contain, in a single case, the primer, the propellant, and the projectile (Figure 1) [8,19,20]. Depending on the percussion method, ammunition types are distinguished in rimfire and centrefire cartridges. The former is used primarily on .22 calibre cartridges and the latter in higher calibres [6].

The weapon’s discharge occurs when the trigger is pulled, followed by a series of mechanical interactions that release the firing pin, which impacts direct or indirectly with the base rim or centre of the cartridge and ignites the primer [21]. This ignition deflagrates the propellant, rapidly generating a large volume of gases that thrust the projectile through its trajectory [19].

The primer currently found in most ammunition consists of a mixture of highly sensitive explosives [6] and other compounds, such as fuel, sensitisers, and oxidisers [19]. In modern primers, the primary explosive is lead styphnate, with barium nitrate as the primary oxidiser [6,19]. The main propellant currently used is smokeless gunpowder (SG), which is characterised by its high burning efficiency, high volume of gas production, low amount of smoke and low amount of debris produced with its deflagration [5,19]. The primary explosive used in SG can be single-based (containing only nitrocellulose (NC)), double-based (containing NC and nitroglycerine (NG)), or triple-based explosive (containing NC, NG, and nitroguanidine (NQ)) [15,16,19,22,23]. The former is the most common type of explosive, whereas the latter is used primarily in military calibre ammunition [16,24]. In addition to the primary explosive, SG can have other components, such as stabilisers, plasticisers, coolants, surface lubricants, flash inhibitors, and sensitisers [5,19,22,25,26,27,28]. These compounds significantly change the chemical and physical properties of the gunpowder and allow it to perform according to the specific purpose for which it was designed [15,25,26,27,28].

GSR, also known as cartridge discharge residue (CDR) or firearm discharge residue (FDR) [12,20], usually consists of burnt and partially burnt fragments from the primer and propellant as well as particles from the cartridge case and the firearm itself [16,20,29,30,31,32]. These residues escape through the firearm’s openings after discharge and can deposit on the hands and clothes of the shooter, clothes of the victim, or nearby surfaces. However, some of the particles stay inside the cartridges and barrel of the gun [33,34,35]. Due to the chemical complexity of the primers and propellants, these residues are complex and heterogeneous mixtures of compounds.

Interestingly, it is precisely the chemical complexity of propellant and primer formulations currently in use, and consequently of the SG and GSR, that allows their discriminative analysis, which assists in the investigation in the context of forensic analysis [16,26,35].

### 2.1. Inorganic Compounds

The inorganic compounds found in gunshot residue (IGSR) derive mainly from the primer, which traditionally contains antimony sulphide, barium nitrate, lead styphnate, lead dioxide, calcium, silicon, and tin [16,18,25,36]. Due to their high molecular masses and low abundance in nature, antimony (Sb), barium (Ba), and lead (Pb) have been the main target of IGSR analysis [20,25,30,31,37,38,39]. In fact, the ASTM 1588-17 standard [40], the Scientific Working Group on Gun Shot Residue (SWGGSR) guidelines [36], and the European Network of Forensic Science Institutes (ENFSI) recommendations [41] focus on the analysis of these elements through scanning electron microscopy with energy-dispersive X-ray (SEM-EDX). This is a non-destructive technique that provides robust morphological and chemical information about the sample particles [32,36,39,40,41,42]. Other methods used for the analysis of IGSR particles, not included in these recommendations, involve atomic absorption spectroscopy (AAS), atomic force microscopy (AFM), proton-induced X-ray emission (PIXE), and neutron activation analysis (NAA). The former provides quantitative and elemental information [13,15,42], whereas AFM allows a very high-resolution picture of the GSR particles at nanoscale dimensions [43]. On the other hand, PIXE is a technique that employs a beam of high-energy particles for elemental analysis [44], and NAA is a highly sensitive approach that focuses on the nucleus of the atoms present in the elements of the sample with quantitative capabilities [16,45]. Moreover, mass spectrometry with inductively coupled plasma (ICP-MS) also proved valuable for IGSR particles analysis [14,16].

Chemical analysis of GSR particles mainly targets inorganic elements, particularly heavy metals, as their co-occurrence is characteristic of GSR. However, with the increase in environmental awareness [46,47], and with the USA [48] and EU [49] pushing to ban the use of lead-containing ammunition, new primer formulations without heavy elements are emerging [39]. In these formulations, heavy elements such as Sb, Ba, and Pb, are now being replaced by copper, zinc, titanium, strontium, iron, nickel, zirconium, steel and aluminium, or other more environmentally friendly organic compounds, such as tetracene, pentaerythritol tetranite (PETN), trinitrotoluene, tetryl, dextrin, diazodinitrophenol, and diazonitrophenol [15,16,18,30,46,47].

Consequently, the IGSR compounds on newer ammunition are less characteristic of gunshot residue since they are naturally present in the environment, and therefore, the probative value of IGSR particles has diminished considerably [12,30,31,50]. To overcome this difficulty, some researchers suggested shifting the target of GSR analysis to organic compounds or to combined analysis of organic and inorganic compounds, which recent studies suggest have higher evidential value [6,16].

### 2.2. Organic Compounds

The organic compounds of GSR are present mainly in gunpowder [15,31], acting as explosives, sensitisers, stabilisers, flash-inhibitors, moderants, coolants, anti-wear additives, and plasticisers [16,28,51,52]. Organic compounds of gunshot residue (OGSR) originate from deflagration of these compounds. In specific cases, OGSR can also originate from the lubricants and other products used to clean firearms barrels [15,31]. Currently, there are over 130 compounds associated with these residues detected and identified [15,16,51], including NC, NG, NQ, nitrobenzene (explosive), methyl centralite (MC—stabiliser and plasticiser), ethyl centralite (EC—stabiliser and plasticiser), akardite I (AK I—stabiliser), akardite II (AK II—stabiliser) 2,4-dinitrotoluene (2,4-DNT—flash suppressor), 2,6-dinitrotoluene (2,6-DNT—flash suppressor), dibutyl phthalate (DBP—plasticiser), diphenylamine (DPA—the primary stabiliser in single-base gunpowder) and its derivates, 2-nitrodiphenylamine (2-nDPA), 4-nitrodiphenylamine (4-nDPA), and N-nitrosodiphenylamine (N-n-DPA) [15,16,18,20,53,54,55,56].

The most common organic compounds in GSR are NG, EC, MC, and DPA and its derivates [54,57], and consequently are usually selected as target analytes in many OGSR studies [28,31,54,57,58,59,60]. Interestingly, many of these additives do not go through the combustion process, which allows their detection on SG and GSR samples [16,61]. Organic compounds can also originate from the primer, especially in newer ammunition models. Several patented primers use organic alternatives to lead styphnate [15,46], such as nitropentene styptic acid, tetrazene, acetogen, polinitropolyphenylether, polinitrophenylether, hexogen, polyvinyl acetate, and red phosphorus stabilised with an acid scavenger and a polymer [15,62].

The analysis of these organic compounds is usually performed through infrared (IR) spectroscopy, Raman spectroscopy, and liquid chromatography (LC) or gas chromatography (GC), often coupled with mass spectrometry (MS) or tandem mass spectrometry (MS/MS) [18,19,25,30], which are described in the following sections.

### 2.3. Potential Sources of SG and GSR Compounds

The widespread existence of GSR-like particles in the environment imposes a serious problem in forensic investigations as they may be confused with GSR particles [32,38]. These GSR-like particles can originate from fireworks [44,63,64], stud guns [38,65], some industrial tools [66], paints [38] or car repair and maintenance products [38]. Police officers’ equipment and vehicles are also susceptible to contaminating suspects and evidence with GSR particles [32,67,68]. However, this contamination was shown to be negligible [68], and the officers’ implementation of simple tasks, such as hand washing or hand sanitising with alcoholic gels, can further mitigate this risk [67].

Several elements commonly found in IGSR, such as Sb, Ba, and Pb, can also be found in the environment. For example, Pb still exists in soils close to major highways and roads due to its long-term use in car fuels. In addition, Pb is still used in solder for plumbing materials, glasses, paints, and battery plates [16]. Sb and derivates (oxidates) exist in some alloys used as fire retardants in cotton and polyester blends [16]. Ba is present in car grease and paints [16]. Nonetheless, despite the multiple sources from which these compounds can originate, their combined presence is a good indication of GSR since there are no environmental sources for this mixture, as mentioned previously [69,70]. Some compounds commonly found in OGSR can also be found in the environment. For example, DPA, the most used stabiliser, can be found on the surface of apples and in outer garments, pesticides, solid rocket fuel, tires, dyes, and veterinary medicine [69,70].

However, due to the removal of the signature elements from the primer, there is a need for the selection of new target analytes. Because the compounds now used as replacement are relatively common in the environment, it is necessary to adopt different analytical methodologies suitable for the determination of these compounds [15,16].

## 3. Analysis of Gunpowder and Gunshot Residue

Chemical analysis has been used to identify GSR particles on suspects, cloths, objects, and surfaces. However, over the last decade, research has been focusing on other applications, such as the estimation of firing or time since discharge [18,52,61] as well as the determination of chemical profiles with discriminative potential to identify manufacturers and even models of ammunition [5,15,18].

### 3.1. Morphological Analysis

The evaluation of the morphologic and physical characteristics, such as the grains’ shape, colour, and size, can be essential in the analysis of both SG and GSR, as it may help to identify or exclude possible manufacturers or ammunition models [33,71]. Furthermore, these characteristics also proved helpful in the determination of shooting distances for GSR particles recovered around entry holes [43].

### 3.2. Chemical Analysis

The first step of chemical analysis is usually the use of presumptive tests. These are simple techniques designed mainly for in-field application, and their use usually precedes more complex chemical analysis. The most common presumptive methods are colour tests, which are inexpensive and simple and consist of rapid sets of procedures [16,19]. However, since most of these tests were designed specifically for field use, they can only provide preliminary results, which considerably decreases their applicability to forensic ballistics [9,15,16]. In this regard, presumptive tests focus almost exclusively on identifying GSR particles in crime scenes, suspects or victims. Furthermore, colour tests can also be used to reveal the dispersion pattern found around the entry wounds of the victims and thus to assist in determining the firing distance [14,16]. Examples of colour tests used in forensic ballistics are the Modified Griees test (MGT), paraffin cast or dermal nitrate tests, Walker test, Marshall and Tewari test, sodium rhodizonate test, Lunge reagent test, Harrison and Gilroy test, and Zincon test. The MGT allows the determination of the total nitrite present in the GSR sample [52], while the paraffin cast or dermal nitrate test detects nitro groups, and the Walker and Marshall/Tewari test detects nitrites. The sodium rhodizonate test detects Pb, and the Lunge reagent test detects NC. Harrison and Gilroy’s test detects Pb, Ba, and Sb, while the Zincon test detects Zn and Ti, which are used in more modern lead-free ammunition [16,20,23].

As mentioned previously, the main target in the chemical analysis of ballistic elements has been, for several years, the elemental analysis of IGSR particles through SEM-EDX, particularly for heavy metals. Methods for the chemical analysis of organic compounds are still in development, and currently, there is a lack of standardised protocols or guidelines. Over the last two decades, researchers have begun to explore the applicability of several other analytical techniques [15,17,18], optimising and exploring the limits of these approaches regarding the chemical analysis of the organic compounds of GSR or SG [27,72,73]. Most of these studies focus on identifying these compounds through chromatographic and spectroscopic methods [74,75,76], the most common methodologies used in forensic chemical analysis. However, other methods have also been explored, such as electrochemical analysis [77] and electrophoretic separation [59,78,79]. Nonetheless, these methods successfully detected and identified only a limited number of compounds [15], which hinders the establishment of guidelines and optimal procedures for this type of analysis. Furthermore, because the performance of each technique may be affected by several different variables, it may impose a case-by-case selection based on the goal of the analysis [15]. The chemical analysis of OGSR compounds should occur as fast as possible and before the analysis of IGSR particles to minimise potential losses during storage or manipulation due to their high volatility [31,80,81]. To further prevent these losses, some authors also suggested the use of corks [82] or aluminium foil [81] to seal the end of the firearm as well as placing the spent cartridges in hermetically sealed vials [83] between sample collection and the subsequent analysis.

To overcome the difficulties in OGSR analysis due to contamination from multiple environmental sources, several researchers suggest the addition of artificial markers in the ammunition manufacturing process [15,32]. These markers would quickly help to unequivocally identify gunpowder and GSR and, possibly, manufacturers and ammunition models if each manufacturer employed different signature mixtures. Nonetheless, using these markers should not interfere with the performance of the ammunition and must be thermally stable, chemically inert, and cheap [84,85,86,87]. Several options were already suggested, including luminescent markers of the lanthanide-organic compounds, such as europium [84,88,89], dysprosium [90], terbium [84,85], or other high-photoluminescence metal tags [87,91]. These substances were shown to maintain luminescent characteristics for up to 30 months, persist on the hands for 9 h, and are very hard to wash off [86]. However, the use of these markers may increase cross-contamination and therefore hinder the interpretation of the acquired data, which diminishes the evidential value of such pieces of evidence [15], rendering its usage controversial and requiring further investigation to be implemented.

#### 3.2.1. Sample Preparation

Sample preparation is essential in various analytical procedures, as it allows the extraction of the target compounds from complex matrices or the removal of interferent analytes [92] that profoundly impact these methods’ analytical performance [92,93]. The most common methods for SG and OGSR sample preparation are solvent extraction, solid-phase microextraction (SPME) and headspace sorptive extraction (HSSE).

##### Solvent Extraction

Solvent extraction uses solvents with different physicochemical properties to extract selected compounds from the sample due to their different partition coefficients [94]. Several solvents have been employed to extract organic compounds from unburnt SG [26,28,95] and spent cartridges [53], including both aqueous solutions [53,96,97] and organic solvents [23,26,28,37,95,97,98,99,100,101]. For example, Dalby et al. [26] used methanol to extract selected compounds for unburnt SG. However, the authors noticed that most analysed samples did not dissolve completely, leading to additional centrifugation and filtration steps before GC-MS analysis [26]. Dichloromethane [97,98], acetonitrile [37,95], ethanol [99], methyl-ethyl-ketone [21,23], and methylene chloride [28,95,100] were also used in the extraction of organic compounds of explosives, SG and GSR samples. Sauzier et al. [101] used acetone to extract compounds from collection devices, choosing it over dichloromethane due to its lower toxicity and ability to dissolve most explosive compounds.

Due to its inefficiency, solvent extraction has seen little use in the chemical analysis of SG or GSR, and it is now being replaced by more modern and efficient extraction methods, such as SPME or HSSE.

##### Solid-Phase Microextraction (SPME)

Solid-phase microextraction (SPME) is a solvent-free extraction technique that allows the extraction and enrichment of volatile and semivolatile organic compounds from the vapour phase of solid, liquid and gas samples at trace or ultra-trace levels [26,27,31,72,92]. The compounds in the vapour phase become adsorbed to the polymeric phase that coats the fused silica fibre [9,15,27,92], which is usually composed of polydimethylsiloxane (PDMS), carboxene (CAR), polyacrylate (PA), divinylbenzene (DVB), or of a mixture of these sorbent phases [73,82,102,103,104]. This extraction process results from an equilibrium established between the sample and extraction phases that is dependent on the characteristics of the sorbent coating and target compounds, the concentration of the analytes, as well as sampling temperature and time [15,26,92]. After the extraction process, the analytes are desorbed from the fibre and transferred to a GC apparatus by thermal desorption in the injection port [92].

SPME is an environmentally friendly extraction technique because of its solvent-free nature (when thermal desorption is employed). It is also relatively cheap, sensitive, and straightforward [26,92]. Furthermore, it eliminates other analytical steps since the compounds can be directly transferred from the sample to the fibre and from the fibre into the injector of the GC [92,103]. Despite all its advantages, SPME also has some limitations, including the inability to extract non-volatile compounds when using headspace extraction and the fact that it is mainly a laboratory-based technique, which limits its in situ use. Moreover, it presents low reproducibility and is more time-consuming when compared to solvent extraction [102,104]. Nevertheless, this extraction technique is recurrent in several forensic areas, including explosives [105] and arson [106] investigations, as well as in forensic ballistics and gunpowder analysis [26,27,82,102,103,104,107].

Andrasko et al. [82,103] were the first to employ SPME in forensic ballistics to estimate the time since discharge of several elements, namely rifles [103], pistols, revolvers [82], spent cartridges, and shotguns [103]. The authors focused on several OGSR compounds using SPME combined with GC-TEA (thermal energy analysis) and GC-FID (flame ionisation detector) [82,103]. Concurrently, Chang et al. [72] studied three standard-loading approaches for the SPME procedure, concluding that in-vial loading of the standard is the most suitable for quantifying trace analysis of selected compounds in SG and GSR. The procedure achieved forensic differentiation among various 9 mm calibre ammunition types by determining the volatile compounds and their relative abundance. In addition, the authors also found this approach suitable for detecting SG in cartridges (factory-made, homemade, or illegally made) or improvised explosive devices and GSR detection [72]. In subsequent studies, Chang et al. evaluated SPME efficiency [107]. SPME fibres coated with 85 µm PA were used to extract the headspace composition of spent cartridges to detect naphthalene, DPA, 2,6-DNT, 2,4-DNT, and DBP, followed by GC-FID analysis. This approach proved successful even after repeated extractions from the same samples (up to seven) [107]. Compared to single extractions from spent cartridges, only minor variations in peak areas for the selected compounds were noted [107]. Dalby and Birkett [26] also studied the effect of fibre composition had on the extraction of OGSR compounds, testing seven different fibres: 65 µm PDMS/DVB, 7 µm PDMS, 30 µm PDMS, 100 µm PDMS, 85 µm CAR/PDMS, 50/30 µm DVB/CAR/PDMS, and 85 µm PA. The former was found to be the most suitable. In addition, the researchers also compared solvent extraction with the SPME extraction of unburned SG compounds, showing that both extraction procedures allowed the detection of the same compounds [26].

Chang et al. [73] further employed a multivariate experimental design to optimise extraction-influencing parameters, including temperature and equilibrium time, to investigate the efficiency of sequential SPME in GSR of spent cartridges. The authors applied the extraction procedure to unburnt SG and successfully extracted volatile compounds that were subsequently identified by GC-FID, showing that the proposed approach can distinguish between ammunition types [73]. Furthermore, Burleson et al. [27] developed a qualitative method for analysing single particles of partially burnt gunpowder using SPME (PDMS, 100 µm film thickness) followed by GC analysis with a nitrogen phosphorus detector (NPD), concluding that this method was suitable for analysing organic components from GSR and had possible forensic applications. The method was also applied to unburnt SG samples, showing considerable differences in the analytical results between unburnt and partially burnt gunpowder.

SPME was also employed in estimating time since discharge by Weyermann et al. [102], who developed a methodology for analysing organic volatiles GSR in spent 9 mm cartridges. Optimisation of the sampling method led them to choose an 85 µm PA-coated SPME fibre, with an extraction time of 40 min at 80 °C [102]. The authors identified 32 organic compounds, selecting 6 to study the potential for dating a gunshot within 32 h after shooting. Benzonitrile, naphthalene, phenol, and 2-ethyl-1-hexanol quickly decreased 2 h after the shot, while 1,2-dicyanobenzene and DPA decreased at a slower rate over 32 h [102].

SPME has emerged as a highly advantageous technique for the enrichment of organic gunshot residue, offering new avenues of research in the field of forensic ballistics [82,102,103,104]. SPME allows for the selective extraction and concentration of OGSR compounds from complex forensic samples. The fibre coating can be tailored to specifically target GSR components, enabling the isolation of relevant compounds while minimizing interference from non-related substances. This selectivity enhances the sensitivity and accuracy of subsequent analysis techniques, improving the overall reliability of forensic ballistics investigations. Furthermore, SPME offers significant advantages in terms of simplicity and cost-effectiveness. The technique requires minimal sample preparation, eliminating the need for labor-intensive procedures and reduces the risk of sample contamination. Moreover, the small-scale nature of SPME makes it amenable to on-site or field applications, allowing for rapid and efficient collection of gunshot residue evidence at crime scenes. This expedites the investigative process, facilitating timely decision-making by forensic experts. Finally, enables the preservation of the original sample, allowing for subsequent analysis using complementary techniques or the re-evaluation of findings as new knowledge emerges. This aspect contributes to the accumulation of a comprehensive database of OGSR profiles, aiding in the establishment of robust forensic protocols and supporting the advancement of forensic ballistics research.

##### Headspace Sorptive Extraction (HSSE)

Headspace sorptive extraction (HSSE) is an extraction technique introduced recently for trace and ultra-trace analysis of volatiles and semivolatile compounds [61,108,109,110]. HSSE has the same theoretical principles as SPME but does not use a thin fibre. Instead, it uses a magnetic stir bar coated with a much larger volume of PDMS or other phases (up to 110 µL) in comparison to SPME (up to 0.5 µL) [61]. Compared to SPME, HSSE has the advantage of better extraction efficiency, higher recovery yields, and the ability to detect more compounds, which directly impact sensitivity and repeatability [9,29,61]. However, to thermally desorb the extracted compounds for GC analysis, a dedicated unit must be installed in the apparatus, seriously limiting the applicability of the methodology.

Using HSSE followed by GC-MS, Gallidabino et al. [61] evaluated the composition and variability of volatile compounds in OGSR from nine handgun ammunition types of two calibres (.357 Magnum and .45 Auto). The authors identified 166 compounds, mainly particles from the propellant, that did not undergo combustion (e.g., additives) [61]. In another study, Gallidabino et al. [29] evaluated the ageing of several OGSR volatiles compounds in two types of .45 Auto ammunition using HSSE, followed by GC-MS analysis. The procedure allowed the detection of 51 GSR compounds from spent cartridges, noting significant differences in their chemical profiles, which the authors believed to correlate with ageing [29]. The authors also claimed that HSSE is more reproducible and effective than SPME and allows the simultaneous analysis of more analytes [29]. Compound-to-compound signal ratios were beneficial in reducing the variability of the ageing curve. This allowed to enhance the time window and were useful for producing data with forensic value [29]. Finally, Gallidabino et al. [108] also developed, optimised, and validated an HSSE/GC-MS method to estimate the time since discharge of 9 mm cartridges, producing data which later allowed the estimation of the time since discharge through multivariate regression analysis [109].

HSSE has shown to be a suitable technique for enriching OGSR from spent cartridges, especially when compared to SPME. However, configurations with different sorbent materials are not easily found on the market, hindering the tailoring effect when compared to SPME. Likewise, the HSSE is less suitable for online thermal desorption since it requires a dedicated unit for this purpose.

#### 3.2.2. Spectroscopy

The main spectroscopic techniques used in SG and GSR analysis are FTIR and Raman spectroscopy [13,111], as shown in Table 1. Infrared (IR) spectroscopy relies on the interaction between infrared radiation and the functional groups in the target molecules, which will vibrate depending on the incident radiation frequency [112,113,114,115]. This methodology is very versatile, allowing the detection, estimation, and determination of organic compounds’ chemical structure in gas, liquid, or solid samples either by absorption, emission, or reflection [13,23,114,115,116,117]. On the other hand, Raman spectroscopy relies on Raman scattering, which is correlated with the polarizability of the electrons in a molecule and provides structural fingerprints that allow molecule identification [50,115,118,119]. Raman spectroscopy allows the analysis of liquid, gas, and solid samples in a non-destructive matter [50]. Both methodologies (FTIR and Raman) can detect and identify IGSR and OGSR, even without heavy metals, in a fast, cost-effective, and non-destructive manner, producing spectra based on the radiation interaction with matter at a given wavenumber [50,74,116,120,121]. Since the methodologies follow different physicochemical principles, they can be used as complementary approaches [74,115].

For example, Brożek-Mucha [122] used IR spectroscopy combined with other techniques, such as optical and scanning electron microscopy and X-ray microanalysis, to evaluate the GSR distribution for close-range shots with a silenced gun, studying it on flesh and cloth. The authors showed that the silencer impacts the distribution and quantity of GSR on surfaces. Mou et al. [43] developed an AFM and an attenuated total reflectance FTIR (ATR-FTIR) procedure to discriminate between manufacturers by analysing GSR particles. Although the authors could not identify specific compounds in GSR samples due to the mixture of debris and other compounds, it was possible to distinguish between manufacturers [43]. Sharma and Lahiri [45] also successfully used FTIR as an alternative to chromatographic methods to analyse OGSR compounds, mainly to detect NG from the suspect’s clothes, and discussed the potential of this technique to determine the shooting distance. In another study, the same authors combined GC-MS, FTIR microscopy, and high-performance thin-layer chromatography (HPTLC) and were successful in characterising and identifying explosives and explosive residues [123]. Bueno and Lednev [74] reported a novel analytical and statistical methodology for the characterisation and identification of GSR. Their analytical approach focused on analysing individual GSR particles by confocal Raman microscopy and ATR-FTIR spectroscopy, as opposed to IGSR-exclusive analysis. This approach is independent of the presence of heavy metals, which increases their usefulness for newer ammunition types [74]. Subsequently, the same authors showed that ATR-FTIR has a high potential for GSR analysis and linking specific suspects with certain ammunition calibres [121]. In later work, as proof of concept, Bueno and Lednev used microscopic ATR-FTIR spectroscopy imaging for the automated detection of IGSR and OGSR particles, focusing on nitrate ester compounds, specifically 2,4-DNT [124].

The first report of Raman spectroscopy applied to this field was published by López-López et al. [23]. The authors were able to detect DPA and its nitration products, MC, EC, and DNT and showed that the obtained spectra had high similarity with those of unburnt SG. This method showed good correlation with GSR and was able to distinguish between GSR-like particles [23]. Subsequently, López-López et al. [111] used Raman with FTIR spectroscopy to compare the profiles of single-, double-, and triple-base SG, proving it useful as a complementary tool for rapid analysis of gunpowder compounds [111]. This study showed that Raman and FTIR had discriminatory potential that depended on the sample constituents. FTIR spectroscopy showed higher discrimination capability between single-base powder with and without DNT and double-base gunpowder, while Raman showed a higher discrimination capability between gunpowder with and without DPA or DNT [111]. Apart from their studies with FTIR, Bueno and Lednev [75] also developed a methodology using Raman microspectroscopic mapping as a proof of concept to detect GSR particles. Subsequently, Abrego et al. [120] developed and implemented a micro-Raman spectroscopy technique to analyse OGSR and a parallel method for IGSR analysis in lead-free ammunition, using scanning-laser ablation inductively-coupled plasma mass spectrometry (SLA-ICPMS). The procedure included a manual microscopical observation of GSR particles followed by Raman spectroscopy to detect OGSR compounds in the samples, which allowed the identification of DPA, its derivates, and centralites [120]. Raman spectroscopy was also used by López-López et al. [23] and Bueno et al. [76] as a complement to SEM-EDX analysis [23,76]. These studies showed a high correlation and identification capacity of ammunition via the obtained spectra [76]. The proposed methodology is suitable for field use due to its fast, non-contact, non-destructive, solventless, and selective nature [23,76]. López-López et al. [99] also applied surface-enhanced Raman spectroscopy to analyse SG and GSR particles after ethanol extraction. The authors detected several compounds, mainly EC, DPA and its derivates. Khandasammy et al. [39] developed a two-step method for detection of OGSR particles with reduced analysis time and potential application sampling large areas. The researchers utilised Raman microspectroscopy to confirm the detection of GSR particle via fast fluorescence mapping, which produces a highly sensitive fluorescence hyperspectral imaging of a sample area, ensuring that the detections were not false positives. According to the authors, this procedure could open the door for the automation of the routine analysis of OGSR compounds. More recently, Khandasammy et al. [125] applied Raman spectroscopy together with laser-induced breakdown spectroscopy (LIBS) to successfully differentiate the OGSR samples from ammunition types of the same calibre and produced by the same manufacturer. LIBS allows rapid, non-destructive, and simultaneous detection of multiple elements present in GSR samples. It requires minimal sample preparation, reducing analysis time and cost. This work showed that LIBS can aid in the identification and characterisation of GSR components accurately. On the other hand, the laser-based nature of LIBS enables remote analysis, making it a promising technique suitable for field investigations.

Raman spectroscopy complements FTIR by offering molecular fingerprinting capabilities [74,115]. It provides detailed information about the vibrational modes of molecular bonds, enabling the identification of specific compounds within GSR [114,115,116,117,118,119]. Raman spectroscopy is particularly useful for differentiating between similar compounds and identifying trace levels of chemicals, even in complex matrices [50,115,118,119]. Moreover, the ability of Raman spectroscopy to probe through transparent substrates makes it advantageous for the analysis of residues on glass surfaces, a common occurrence in forensic ballistics cases. Recent developments in both FTIR and Raman spectroscopy have opened new avenues for research in forensic ballistics [50,74,116,120,121]. Advances in instrumentation, such as portable and handheld devices, now allow on-site analysis, reducing the need for sample transportation and facilitating rapid decision-making in crime scene investigations. Moreover, the integration of imaging techniques with spectroscopic methods offers enhanced spatial resolution, enabling the visualization and mapping of GSR distribution patterns. This development allows for a more comprehensive understanding of shooting incidents, including the determination of shooting angles and the identification of multiple shooting events.

**Table 1 molecules-28-05550-t001:** Summary of some spectroscopic techniques employed in GSR analysis.

Technique	Objectives	Target Analytes	Matrix	Conclusions	Ref.
Optical and scanning electron microscopy; X-ray microanalysis; infrared spectroscopy	Evaluation of the GSR distribution for close-range shots with a silenced gun	GSR	Cotton cloth and fresh porcine skin	Attaching a silencer to the studied weapon significantly modifies the distribution and amount of GSR on the tested surfaces	[122]
ATR-FTIR and AFM	Discrimination between different manufacturers by analysing GSR particles	GSR	Polyethylene and aluminium foil sheets	Identifying specific compounds using these techniques was not possible, but different bands on FTIR spectra may help to identify the manufacturer	[43]
FTIR microscopy	Detection and estimation of NG and other GSR on suspects’ hands and clothes	OGSR (mainly NG)	Cloth	A promising method to estimate the shooting distance	[45]
Raman	Identification of OGSR using Raman (first study)	MC, EC, DNT, and DPA and its nitration products,	Unburnt gunpowder and GSR	Raman was a helpful screening tool for GSR, and to distinguish it from other particles; establish a correlation between intact and burnt gunpowder	[23]
Raman and FTIR	Comparison of profiles obtained from both techniques. Discriminate and identify different gunpowder	OGSR	Gunpowder solutions (in methyl ethyl ketone)	Combining FTIR and Raman spectroscopy with discriminant analysis proved to be a valuable tool for the classification and the possible identification of unknown samples of gunpowder	[111]
Raman and FTIR	Development of a new analytical and statistical approach to GSR analysis	OGSR	-	Both spectroscopic techniques provide complementary information	[74]
Microscopic ATR-FTIR spectroscopy imaging	Automated detection of IGSR and OGSR particles using automatic visual scanning	GSR	Cloth (followed by tape lifting)	New automatic method to detect macro and microscopic particles and determine the “vibrational fingerprints”	[124]
Raman microspectroscopic	Chemical mapping and automated GSR particles detection	GSR	Cloth (followed by tape lifting)	Development of a procedure which is not dependent on heavier metals in GSR	[75]
ATR-FTIR	Establish a link between evidence and suspects	GSR	-	High potential for GSR analysis and linking specific suspects with certain ammunition calibres	[121]
Raman	Spectroscopically characterise and statistically explore differences in the Raman spectra of GSR of different calibre weapons	GSR	Cloth	High correlation and identification capacity of ammunition via the obtained spectra	[76]
Surface-enhanced Raman scattering (SERS)	Development of a new analytical procedure for fast and sensitive analysis of GSR	GSR	Gunpowder and GSR solutions	Detection of several compounds, mainly EC, DPA, and its derivates	[99]
Micro-Raman spectroscopy; SLA-ICPMS	Detection and identification of GSR compounds (IGSR and OGSR)	OGSR (micro-Raman); IGSR (SLA-ICPMS)	Tape-lift (modified)	Capable of detecting GSR from shooters’ hands	[120]
Raman and LIBS	Differentiate OGSR samples from ammunition types of the same manufacturer	OGSR		Aid in the identification and characterisation of GSR components accurately	[125]

#### 3.2.3. Chromatography

Chromatography, a powerful analytical technique, plays a pivotal role in the analysis of GSR for forensic ballistics investigations. Chromatography offers several advantages that make it an indispensable tool for GSR analysis, including its high sensitivity, selectivity, and ability to separate complex mixtures [126].

##### Liquid Chromatography

Liquid chromatography (LC) is a highly sensitive and reproducible technique commonly used in forensic sciences. It is suited for separating non-volatile, semivolatile, and thermolabile compounds and can be combined and/or coupled with a wide range of detectors, conferring high flexibility to the technique [14,126]. However, LC has several limitations, such as extensive sample preparation for some applications, large consumption of organic solvents, and its destructive nature with regards to the probative value of the sample (depending on the detector used) [15,18]. Nonetheless, several works have already explored the use of LC coupled with MS [18,37,56,57,68], diode array detector (DAD) [127,128], and ultraviolet (UV) [16,54] detectors for forensic ballistics (Table 2).

**Table 2 molecules-28-05550-t002:** Summary of some LC techniques employed in GSR analysis.

Technique	Objectives	Target Analytes	Matrix	Conclusions	Ref.
LC-MS/MS; SEM-EDX	Separation and detection of both IGSR and OGSR in the same sample	GSR	GSR solution (in methanol and acetonitrile)	Development and validation of the methodology	[37]
UPLC/MS/MS	Separation and detection of OGSR compounds	Organic components from SG	Gunpowder solutions (in methylene chloride)	Separation and identification of 21 OGSR compounds. According to the authors, this procedure allows the differentiation between brands and lots by analysing the compositional differences	[28]
UHPLC-UV	Separation and identification of 32 target OGSR compounds, with the aid of Artificial Neural Networks (ANN)	32 OGSR target compounds	Gunpowder solutions (in dichloromethane); GSR solutions (in MTBE, after hand swab)	Separation and identification of 32 OGSR, faster and with lower LOD, thanks to ANN optimisation	[54]
HPLC	Prediction of the age of gunpowder, with the aid of statistical models	Derivates of DPA, mainly N-nitroso-DPA	Gunpowder solutions (in methanol)	Successfully determined the age of gunpowder samples using multiple linear regression with a square root transformation model	[22]
LC-MS/MS	Development of methodologies for the analysis of OGSR to determine their application in chemical ballistic	OGSR	GSR solutions (in isopropyl/water, after hand swab)	New protocols for sample collection and preparation and analysis procedure for OGSR	[57]
UHPLC-MS	Comparison of the efficiency of various sampling materials in collecting OGSR	OGSR	GSR solutions (in methanol, after hand swab); Gunpowder solutions (in methanol)	Modern instrumentation allied with efficient sample preparation makes it easier to detect and identify OGSR from discharged material, even a few hours after discharge	[56]

Minzière et al. [25] used ultra-high-performance LC (UHPLC) coupled with tandem mass spectrometry (MS/MS) to evaluate the simultaneous analysis of ISGR and OGSR. The authors evaluated the latter by analysing eight organic compounds commonly found in gunpowder (NG, DPA, AK II, EC, 4-nDPA, 2-nDPA, 2,4-DNT, and N-n-DPA) to compare three sampling procedures. With a similar goal, Taudte et al. [30] used UHPLC-UV-MS/MS to develop and compare collection techniques for both OGSR and IGSR. The extraction protocol for OGSR analysis consisted of liquid extraction with acetone and preconcentration before instrumental analysis, which potentially increases the probative value of GSR when using the developed sampling method in combination with SEM-EDX and UHPLC methodologies [30]. Feeney et al. [37] developed and validated an LC-MS/MS method to separate and detect both IGSR (used SEM-EDX) and OGSR in the same sample, increasing the confidence of the obtained chemical profile. Thomas et al. [28] developed and employed a fast UPLC-MS/MS methodology to separate and identify analytes in GSR samples. The authors were successful in identifying 21 OGSR compounds. The authors were able to differentiate between brands and lots by analysing the compositional differences [28]. Taudte et al. [54] optimised a UHPLC-UV method to separate and identify 32 OGSR compounds with lower limit of detection (LOD) than previously reported. This increased sensitivity allowed for the detection of trace compounds, significantly augmenting method applicability.

In a different approach, López-López et al. [22] used HPLC-DAD and statistical models to predict the age of SG. The authors chose HPLC-DAD with an isocratic water/acetonitrile mobile phase to avoid the thermal degradation of N-nitroso-DPA when analysed by GC. The authors focused on derivates of DPA, mainly N-nitroso-DPA, since the DPA nitration process is stable during the ageing of SG [22,55,129]. Also focusing on the additives of gunpowder, Laza et al. [57] proposed an LC-MS/MS method for analysing OGSR stabilisers, such as DPA.

With a different objective, Gassner and Weyermann [56] developed a UHPLC-MS methodology to compare the efficiency of various sampling materials for collecting OGSR. The authors showed that modern instrumentation and an efficient sample preparation technique facilitated detecting and identifying OGSR from discharged material a few hours after discharge [56].

Liquid chromatography (LC) is a valuable tool for analysing GSR in forensic ballistics due to its excellent separation capabilities, sensitive detection methods, flexibility in sample preparation, and automation potential [126], enabling accurate identification and quantification of GSR components for criminal investigations.

##### Gas Chromatography

Gas chromatography (GC) is among the most used techniques in the forensic science, especially when coupled with mass spectroscopy (MS) [51,93,100,130]. Within the field of forensic ballistics, GC was already employed in the analysis of unburnt SG [26,104,131] and GSR [27,102] (Table 3). This method presents many advantages, such as low analysis time, low detection limits (at the nanogram level), and high selectivity and sensitivity [14,16,93]. However, GC is a destructive technique that can only be used to analyse volatile and semivolatile compounds, thus excluding NC and NG [14,15,16,126]. Moreover, NC may decrease the GC column’s lifetime [16]. In addition, thermally unstable compounds, such as nitrate esters (e.g., pentaerythritol tetranitrate), can decompose during the analysis [15]. Furthermore, this technique cannot analyse both inorganic and organic compounds, although this can be circumvented through sampling or extraction techniques that could allow separate analysis of these compounds [15].

GC has been coupled with several detectors to analyse SG and GSR, including a flame ionisation detector (FID) [93,103,132], mass spectroscopy (MS) [31,51,93,100,130,133], ion mobility spectroscopy (IMS) [100,134], a nitrogen phosphorous detector (NPD) [26], and thermal energy analysis (TEA) [100,103].

**Table 3 molecules-28-05550-t003:** Summary of some GC techniques employed in GSR analysis.

Technique	Objectives	Target Analytes	Matrix	Conclusions	Ref.
SPME/GC-FID	Estimate the time since discharge and environmental effects on the estimation based on the degradation of organic compounds on GSR	Naphthalene; 2,6-DNT; 2,4-DNT; DPA; DBP	Spent cartridges	Successfully detected the organic compounds in the cartridge up to 14 days after firing; reliable determination of time since discharged based on DPA, DBP and naphthalene	[132]
SPME/GC-MS	Determination of the time since discharge of spent cartridges	OGSR	Spent cartridges	Detection of 32 OSGR in spent cartridges; DPA and 1,2-dicyanobenzes decrease the slowest over 32 h	[102]
SPME/GC-MS; SEM-EDX	Obtain chemical profiles of single GSR samples collected from the shooter’s hands	OGSR (GC) and IGSR (SEM-EDX)	Unburnt gunpowder and GSR	Successfully determined the chemical profile of samples, using the two techniques combined	[31]
SPME/GC-FID	Optimisation of an SPME procedure and determination of the viability of multiple extractions	OGSR	Spent cartridges	Spent cartridges can be analysed repeatedly and non-destructively (if appropriately sealed)	[107]
SPEM/GC-MS	Determination of the most suitable SPME fibre for extracting OGSR compounds	DPA; 4-NDPA; EC; NG; DBP	Unburnt gunpowder	By comparing the average peak areas of the compounds, the most suitable fibre type was determined to be the 65 µm PDMS/DVB	[26]
SPME/GC-NPD	Development of an analytical method to analyse a single particle of partially burnt gunpowder	OGSR	A single particle of partially burnt gunpowder; unburnt gunpowder	Successfully detected organic compounds in the sample	[27]
HSSE/GC-MS	Evaluate the composition and variability of volatile compounds in OGSR in handgun ammunition	OGSR	GSR (after HSSE extraction)	Identification of 166 compounds, most being additives of gunpowder	[61]
HSSE/GC-MS	Study of the ageing of several OGSR volatiles compounds	OGSR	Spent cartridges	Detection of 51 OGSR compounds, which presented noticeable ageing profiles	[29]

GC can also be combined with thermal desorption (TD/GC), mainly when using HSSE or SPME [29]. For example, Chang et al. [132] developed an SPME/GC-FID procedure to analyse discharged cartridges and establish the time since discharge based on the degradation of organic compounds. The quantity of these analytes was also evaluated as a function of time to estimate GSR persistence upon exposure to environmental factors [132]. The results show that the authors successfully determined the time since discharge up to 14 days after firing, using several constituents of SG, namely naphthalene, 2,6-DNT, 2,4-DNT, DPA, and DBP. Environmental factors, such as sunlight exposure to discharged materials, were also studied [132]. Weyermann et al. [102] also developed a SPME/GC-MS method with similar goals. This approach detected 32 OSGR compounds in 9 mm cartridges after discharge over 32 h. The results showed that the concentration of some compounds quickly decreased after the discharge (e.g., phenol, 2-ethyl-1-hexanol, and naphthalene), while others could still be detected 32 h after shooting (e.g., DPA and 1,2-dicyanobenzene) [102].

Goudsmits et al. [31] developed a methodology of analysing IGSR and OGSR collected from a suspect’s hands. This methodology included an SPME/GC-MS analysis of OGSR compounds and an SEM-EDX analysis of IGSR particles. This combined analysis allowed the authors to obtain a complete chemical profile from the GSR samples. Tarifa and Almirall [133] also combined two different rapid methodologies to characterise OGSR and IGSR detected on a suspect’s hand. This approach consisted of a capillary microextraction of volatiles (CMV) followed by GC-MS and LIBS [133].

GC offers significant advantages for analysing GSR in forensic ballistics. It provides sensitive and selective detection of trace amounts of OGSR and volatile organic compounds (VOCs). By coupling GC with mass spectrometry (GC-MS), specific compounds can be identified accurately, minimizing false-positive results. GC allows for comprehensive analysis of a wide range of GSR components, aiding in the determination of shooting distance, firing frequency, and firearm correlations. Its speed, efficiency, and ability to create databases for comparative analysis enhance the accuracy of forensic investigations.

### 3.3. Emerging Applications

The adoption of new chemical analytical techniques for detecting and identifying organic compounds in SG and GSR usually results in the acquisition of a large amount of data. Most of these data points are not useful for the comparison of questioned samples within forensic ballistics. However, this opens the possibility for the exploration of novel avenues of approach that may allow responding to other vital questions within the field of forensic ballistics. In fact, over the last decade, several researchers have explored new applications for this data, taking advantage of the most recent statistical and machine learning tools currently available for model development [13] and of modern computational database systems for efficient storage and data access [135]. Within the forensic sciences, several research works have already proved the usefulness of these chemometric tools, as visible in Table 4. These helped in the clustering, interpretation, and optimisation of several analytical procedures, particularly those that rely on visual comparisons of spectra, chromatograms or other similar data [116,117,136,137,138,139].

Chemometric tools, which cover a wide range of applications, can be divided into three main categories: (i) pattern recognition, which can be supervised or unsupervised, and is oriented toward the automated recognition of relations in datasets; (ii) regression methods, which predict sample characteristics quantitively; and (iii) experimental design, used to optimise chemical procedures [135].

Within forensic ballistics, pattern recognition and regression methods have already been explored to analyse SG and GSR chemical data. These methods include hierarchical clustering analysis (HCA), principal components analysis (PCA), partial least squares (PLS), PLS associated with discriminant analysis (PLS-DA), k-nearest neighbour (KNN), multivariate adaptive regression splines (MARS), random forests (RF), artificial neural networks (ANN), and support vector machines (SVM) [13,50,54,134,140,141,142,143,144,145,146,147,148]. HCA is a clustering method that groups objects iteratively, agglomerating or dividing clusters in each cycle based on objects’ similarity for a determined distance measure [135,143]. PCA is an unsupervised data reduction methodology that transforms data linearly, creating a new set of orthogonal variables that successively maximise the variance present in the data [135,142,149]. PLS is a supervised regression methodology that, similarly to PCA, transforms data linearly by projecting the variables into a new space, after which a linear regression model is computed [140]. PLS-DA is a supervised method that extends PLS to classification tasks [144,145]. KNN is a supervised pattern recognition based on the distance between known and unknown objects, trying to group the unknown objects with their closest k-neighbours to attribute a class to the object [150]. ANN are predictive models suitable for classification and regression formed by multiple layers of interconnected perceptrons, similar to biological neurons, weighing each input and producing an output [54,135,148]. MARS is a non-parametric regression analysis technique that adapts linear regression methods to non-linear interactions between variables [146,147]. RF are an ensemble of independent decision trees, in which each tree comprises groups of nodes representing a test on a particular object attribute [135]. SVM are supervised method for classification that separates classes based on the maximum distance between them in a hyperplane, used when known classes cannot be linearly separated [135,151].

Making use of some of these statistical analysis tools, Reese et al. [152] developed a non-targeted approach for characterising unburnt SG and OGSR. The authors applied their method to various ammunition from different manufacturers, calibres, and ages based on the chemical profile obtained. From this procedure, using LC/time-of-flight (TOF)-MS, a statistical analysis was conducted using PCA and HCA to discriminate unburnt gunpowder based on chemical composition and to establish a correspondence between unburnt gunpowder and OGSR compounds [152]. Although good results were obtained, these tools, as any other machine learning tools, must be used with care since their outcomes are highly reliant on the quality and representativeness of the data. On the other hand, many of these tools may struggle to distinguish between samples with minor compositional variations, particularly when the differences fall within the limits of the analytical techniques employed.

**Table 4 molecules-28-05550-t004:** Summary of some emerging applications employed in GSR analysis.

Statistical Method	Experimental Procedure	Target Analytes	Conclusions	Ref.
PCA and HCA	LC-TOF/MS	SG and OSGR	Discrimination of SG based on the chemical composition by matching SG’s organic compounds to OGSR	[152]
Spearman’s correlation test	HPLC and micellar electrokinetic	SG	The comparison of both techniques showed slightly different results and complementary potential	[127]
Database and analysis of the statistical impact	Raman and FTIR	GSR	Creation of a database with combined FTIR and Raman spectra; determination of the different impacts that these techniques had on a chemometric model	[74]
PLS and SVM	NIR Raman	GSR	Successful discrimination and identification of GSR particles	[76]
LR	SPME/GC-MS	GSR	Creation of a logical approach to determining the time since discharge, with a successful application to a hypothetical scenario	[33]
Pairwise log ratio normalisation combined with RF and PLS regression	HSSE/GC-MS	GSR	Estimation of time since shooting on spent cartridges	[109]
LR		GSR	Evaluation of judgment and conclusions of forensic experts in identification ballistics, determining their results to have high sensitivity and specificity	[7]
PCA and PLS-DA	ATR-FTIR	GSR	Successful discrimination of ammunition calibre	[121]
ANN	UHPLC-UV	OGSR	Prediction of retention time of 32 OGSR compounds during method optimisation	[54]
ANN	IMS	GSR	Differentiation of particles collected by hand swabs, discriminating between shooters and non-shooters	[148]

Bueno and Lednev [74] also reported a novel analytical and statistical methodology for analysing GSR based on the Raman and FTIR spectra dataset. The researchers found that these two analytical techniques contributed differently to the chemometric model. However, the authors also showed this to be a robust approach in forensic investigations for ruling out particular firearm ammunition [74]. The same authors [76] also studied the potential of near-infrared (NIR) Raman to discriminate and identify particles of GSR using PLS correlation analysis and SVM.

With a different objective, Cascio et al. [128] used Spearman’s correlation test to compare HPLC and micellar electrokinetic capillary chromatography in their capability to analyse the organic compounds of SG. The results were slightly different with each technique, and Spearman’s correlation test showed a good relationship between the different separations’ patterns, pointing to the possibility of complementary approaches [128]. Gallidabino et al. [33] based their study on a hypothetical SPME/GC analysis of spent cartridges to create a logical approach to estimate their time since discharge. The researchers used likelihood ratios to develop a probabilistic model and applied it to a hypothetical scenario, claiming that all the required parameters for such a model could be easily estimated from seized material [33]. In another approach, Gallidabino et al. [109] used chemometrics to estimate the time since shooting from spent cartridges based on their previously optimised and validated HSSE/GC-MS methodology. The authors tested PLS, MARS, ANN, RF, KNN, and SVM regression algorithms and found the most suitable models to be those trained with RF and PLS [109].

Bueno et al. [121] combined ATR-FTIR with statistical analysis, PCA, and PLS-DA. The PCA analysis revealed that samples from the same calibre were grouped. At the same time, PLS-DA was able to differentiate between the three calibres analysed in the study. The authors did not discuss the reason for obtaining better results using this system [121]. However, it is known that PLS-DA can effectively reduce the dimensionality of the data by identifying the most relevant set variables for the problem in question. This is particularly advantageous for gunshot residue analysis, as it enables the identification of key elemental or spectral characteristics that distinguish various types of residues.

Taudte et al. [54] applied ANN under a predictive data-processing program to predict the retention time of 32 OGSR compounds analysed via UHPLC-UV using several gradients, allowing the detection of all of the studied compounds. Bell and Seitzinger [148] have also used ANN to differentiate samples from shooters and non-shooters based on ion mobility spectra obtained from hand swabs.

Regarding visual comparison of firearm-related elements, Mattijssen et al. [7] used computer-based methods to evaluate the validity and reliability of judgments and conclusions of forensic experts. The procedure is based on data acquisition in 2D and 3D, followed by data pre-processing and comparing striation patterns to calculate the likelihood ratio [7]. The study showed the high sensitivity and specificity of the conclusions presented by the examiners. Still, those conclusions seem slightly less proficient at relating samples from the same source and better at distinguishing samples from a different source when compared with the computer-based method [7].

These chemometric tools and applications offer significant potential for advancing forensic ballistics research by enabling more advanced and efficient analysis of GSR data. They have the capability to contribute to the establishment of comprehensive databases and classification systems for GSR samples, assist in the identification of firearm characteristics, facilitate the linking of evidence to specific firearms or crime scenes, and provide valuable insights into the factors influencing GSR characteristics. Moreover, these tools can aid in the interpretation of forensic evidence, enhance the accuracy of forensic examinations, and potentially pave the way for the development of automated or semi-automated GSR analysis systems. Ongoing developments in this field aim to refine and optimise these techniques, integrate multiple methods to ensure more robust analysis, and explore the application of emerging chemometric tools.

## 4. Summary

Recent legislative proposals in several countries indicate a change in the paradigm that will lead to the removal of heavier metals from ammunition. These changes eliminate the inorganic compound mixture characteristic of GSR and may force researchers to shift the focus of their analysis to OGSR compounds. Therefore, methods more suitable for analysing OGSR compounds, such as chromatography and spectroscopy, are being developed.

Several studies have reported the successful use of alternative methods to detect, identify, and quantify OGSR compounds. Nonetheless, some researchers point to better efficiency when combining both OGSR compounds and IGSR particles analysis. For this reason, combination of the standard analysis method of IGSR particles (SEM-EDX) with chromatographic or spectroscopy analysis of OGSR compounds could be a suitable avenue for future research for forensic laboratories dealing with GSR analysis. The combination of two techniques with different target analytes—or a single technique with the capability of analysing both organic and inorganic GSR, such as Raman or FTIR spectroscopy [50,74,116,120,121]—could help to accomplish this goal. The results of the combined analysis of organic and inorganic residues could be used, in the best scenario, to correlate a sample with reference material, aiding in the investigation of a crime in which firearms were used.

Up to now, no standard analytical methodology for OGSR compounds has been implemented. SPME followed by GC-MS is one of the most promising methodologies for achieving standardisation, as it has consistently yielded superior results according to numerous authors [102,107,109,132]. In addition, a standardized method for data analysis must also be defined. Chemometrics showed that it can be a suitable method of studying the data and creating databases. This information can potentially be used with several statistical procedures [13,54,135,140,141,142,143,144,145,146,147,148] to compare and identify samples based on those databases.

These alternative analytical methods can be extremely helpful in forensic ballistics, particularly for firearm-related elements found in crime scenes in which the conventional forensic ballistic visual comparison cannot be used. In addition, these methods produce a considerable number of data points that have driven, and currently are driving, the exploration of new approaches to answer other vital questions within this field. These include the development of predictive models to determine time since shooting and firing distances and to identify ammunition’s manufacturer and its model. These methods can take advantage of the most recent statistical and machine learning tools currently available for model development and of modern computational database systems for efficient storage and data access.

## Figures and Tables

**Figure 1 molecules-28-05550-f001:**
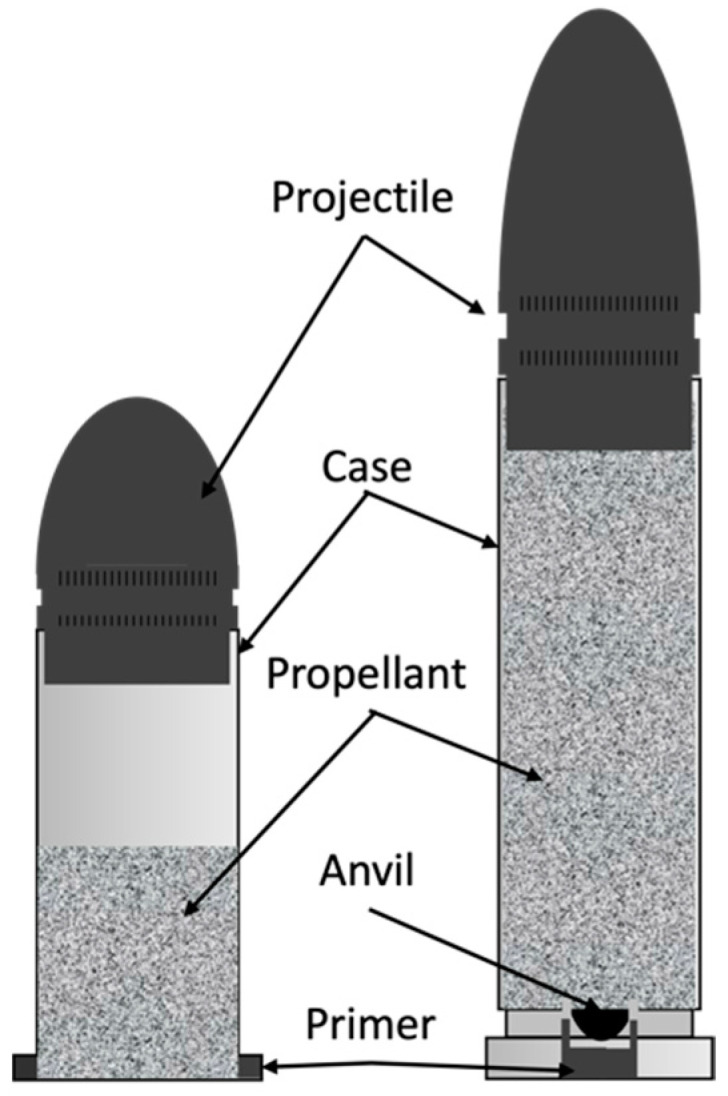
Schematical representation of a standard “rimfire” .22 cartridge on the left and a standard “centrefire” ammunition on the right.

## Data Availability

Data sharing not applicable.

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
