# Peer review of "Chemical Analysis of Gunpowder and Gunshot Residues"

_molecules, 2023, doi:10.3390/molecules28145550_

Round 1
Reviewer 1 Report
This is a timely review article. The law enforcement community is in a great need for more specific and informative methods for gunshot residue detection and analysis. The article is well written and can be recommended for publication after a minor revision. My main suggestion is to expand Spectroscopy section by including the discussion of a novel two-step method for the detection and identification of GSR based on the fast fluorescence imaging followed by Raman microspectroscopic identification (DOI: 10.1021/acs.analchem.9b02306). Readers will also benefit from having references on recent comprehensive reviews on the subject of this article including a general review of emerging forensic methods (DOI: 10.1021/acs.analchem.8b04704) and GSR specifically (doi.org/10.1016/j.trac.2017.12.003).
Specific Comments
· Abstract (Line 14): “such as projectiles and casings”- change “casings” to “cartridge cases”
· Introduction (Lines 33-34): “being one of the primary concerns of law-enforcing agencies across the globe”- change “law-enforcing” to “law enforcement”
· Introduction (Line 35): “For example, in 2019…”- why choose 2019? There should be information for 2020-2022.
· Introduction (Lines 54-55): “Whenever this physical analysis is not possible, the chemical analysis of gunshot residues (GSR)…”- residue should not be plural, should be the “analysis of gunshot residue”
· Main components of gunpowder and gunshot residues (Line 101): “any surface in its vicinity”- be more specific here: on the hands of the shooter, clothes of the shooter, clothes of the victim, nearby surfaces, etc.
· Inorganic Compounds (Line 112): “The inorganic compounds found in gunshot residues (IGSR)”- same issue with plurality
· Inorganic Compounds (Lines 113-114): “lead styphnate and dioxide”- change it to “lead styphnate and lead dioxide”
· Inorganic Compounds (Lines 114-115): “Due to their high molecular mass and low abundance in nature, Sb, Ba and Pb…”- these compounds should be identified as antimony (Sb), barium (Ba), and lead (Pb) when mentioned for the first time.
· Inorganic Compounds (Lines 135-136): “In these formulations heavy elements such as Pb, Sb, and Ba”- they changed the order of the common heavy metals; they should keep the order consistent with before: Sb, Ba and Pb
· Potential sources of SG and GSR compounds (Line 182): “vehicle brakes linings [63,64]”- Raman spectroscopy has been reported (doi.org/10.1007/S00216-018-1359-1) does NOT show false positives due to vehicle brakes linings.
· Potential sources of SG and GSR compounds (Line 189): “In addition, Pb it is still used in”- remove the “it”
· Potential sources of SG and GSR compounds (Lines 198-202) “However, due to the removal of the signature-elements from the primer, and subsequently from the GSR particles and because the compounds used as replacement are common in the environment, the selection of new target analytes is a real necessity. Imposing also the adoption of with different analytical methodologies suitable for the analysis of these compounds [15,16]”- this is too long and wordy; also “the adoption of with different…”, the ”with” needs to be removed
· Analysis of gunpowder and gunshot residues (Line 206): “Chemical analysis has been used to identify GSR particles in suspects…”- the word “in” is an odd word to use, I think “on” would be more appropriate; it is known that GSR can penetrate into some clothing, but for the most part, “on” would be most appropriate
· Analysis of gunpowder and gunshot residues (Lines 207-208): “However, over the last decade, research has been focusing on the expanding the applications of the chemical analysis”- in this sentence, remove the “the” in “the applications”- it has two redundant “the” in the sentence
· Analysis of gunpowder and gunshot residues (Lines 228-229): “they can only provide indicative results”- I don’t understand the use of “indicative” here. Maybe, “preliminary” is a better word here
· Analysis of gunpowder and gunshot residues (Line 297): “methyl-ethyl-ketone [22]”- consider this work here (doi.org/10.3390/chemosensors11010011)
· Analysis of gunpowder and gunshot residues (Lines 335-336): “The authors focused on several OGSR using SPME”- I would add the word compounds after OGSR
· Analysis of gunpowder and gunshot residues (Line 386): “to thermally desorbed the extracted compounds for GC analyses…”- “desorbed” should not be past tense, change to “desorb”
· Analysis of gunpowder and gunshot residues (Lines 414-417): “Both methodologies can detect and identify IGSR and OGSR, even without heavy metals, in a fast, cost-effective, and non-destructive manner, producing IR spectra based on the radiation interaction with matter at a given wavenumber [72,114,118,119]”- the authors referenced both IR and Raman works, but then only said IR spectra; I would just remove “IR”
· Analysis of gunpowder and gunshot residues (Line 445): “They showed that the obtained spectrums had high similarity”- not “spectrums;” replace with “spectra”
· Analysis of gunpowder and gunshot residues (Line 461): “followed by a Raman spectroscopy to detect OGSR in the samples”- remove the “a” in the sentence
· Table 1, Raman and FTIR, Objectives “Comparison of IR profiles obtained from both…”- remove “IR”
· Analysis of gunpowder and gunshot residues (Lines 482-483): “The signal produced by these detectors is directly proportional to the compound's concentration on the sample”- consider changing “on” to “of”
· Analysis of gunpowder and gunshot residues (Lines 484-487): “Chromatography varies based on its stationary and mobile phases, being the most common in forensic laboratories those with a solid stationary phase and either a liquid mobile phase (liquid chromatography) or a gas mobile phase (gas chromatography)”- this a long run-on sentence that is oddly worded. How about… “Chromatography varies based on its stationary and mobile phases. In forensic science, there is a solid stationary phase with either a liquid mobile phase (liquid chromatography) or a gas mobile phase (gas chromatography)”
· Analysis of gunpowder and gunshot residues (Lines 535-536): “Gas Chromatography (GC) is among the most used techniques in the forensic context, in particular when coupled with mass spectroscopy (MS) [47,91,98,127].”- sounds a little odd; How about… “Gas Chromatography (GC) is among the most used technique in forensic science, especially when coupled with mass spectroscopy (MS)”
· Analysis of gunpowder and gunshot residues (Lines 560-561): “This approach detected 32 OSGR in 9 mm cartridges after discharge over 32 hours.”- there should be a word following OGSR; I assume the authors mean OGSR compounds
· Analysis of gunpowder and gunshot residues (Lines 570-572): “This approach consisted of a capillary microextraction of volatiles (CMV) followed by GC-MS and laser-induced breakdown spectroscopy (LIBS) 571[130]” – It is important here to add the discussion of Raman and LIBS combination for the detection and analysis of GSR (doi.org/10.1016/j.saa.2023.122316).
· Summary (Lines 689-690): “Allied to this, a standardized method for data analysis must also be defined” – the word “allied” is an odd choice; I would remove it entirely and say, “A standardized method for data analysis must also be defined”
· Summary (Lines 694-695): “These alternative analytical methods can be extremely helpful in the forensic ballistics”- remove “the”
· Summary (Lines 698-703): “These include the development of predictive models to determine time since shooting and firing distances and to identifying ammunition’s manufacturer and its model, all taking advantage of the most recent statistical and machine learning tools currently available for model development and of modern computational database systems for the efficient storage and access to this data.”- this is one LONG sentence and has some grammar issues. How about… “These include the development of predictive models to determine time since shooting, firing distances, and to identify firearm manufacturer and its caliber. These methods can take advantage of the most recent statistical and machine learning tools currently available for model development and of modern computational database systems for the efficient storage and access to this data.”
Minor editing of English language is required
Author Response
Response to reviewer #1 comments and suggestions
Overall comments
This is a timely review article. The law enforcement community is in a great need for more specific and informative methods for gunshot residue detection and analysis. The article is well written and can be recommended for publication after a minor revision. My main suggestion is to expand Spectroscopy section by including the discussion of a novel two-step method for the detection and identification of GSR based on the fast fluorescence imaging followed by Raman microspectroscopic identification (DOI: 10.1021/acs.analchem.9b02306). Readers will also benefit from having references on recent comprehensive reviews on the subject of this article including a general review of emerging forensic methods (DOI: 10.1021/acs.analchem.8b04704) and GSR specifically (doi.org/10.1016/j.trac.2017.12.003).
We would like to express our gratitude to the reviewer for the valuable comments and suggestions, which have significantly contributed to the improvement of the manuscript. We have expanded the “Spectroscopy” section as recommended, specifically addressing the integration of Raman and LIBS techniques for the detection and analysis of gunshot residue (GSR) (Lines 513-528, spectroscopy section). Additionally, we have incorporated updated data pertaining to firearm-related fatalities. We also implemented the requested changes in English editing across the entire manuscript, with particular emphasis on the revisions explicitly indicated by the reviewer. To address the specific concerns and comments raised by the reviewers in a comprehensive manner, we have provided a detailed point-by-point response below.
Specific Comments
Abstract (Line 14): “such as projectiles and casings”- change “casings” to “cartridge cases”
The word “casings” was replaced by “cartridge cases” (Line 14).
Introduction (Lines 33-34): “being one of the primary concerns of law-enforcing agencies across the globe”- change “law-enforcing” to “law enforcement”
The term “law-enforcing” was replaced “law enforcement” (Lines 35-35).
Introduction (Line 35): “For example, in 2019…”- why choose 2019? There should be information for 2020-2022.
After checking additional sources, in particular https://worldpopulationreview.com/country-rankings/gun-deaths-by-country We found that the displayed data in the manuscript is current. That particular website cites the same sources as we do in our manuscript.
Introduction (Lines 54-55): “Whenever this physical analysis is not possible, the chemical analysis of gunshot residues (GSR)…”- residue should not be plural, should be the “analysis of gunshot residue”
The word “residues” was replaced by “residue” (Line 56), and in other instances.
Main components of gunpowder and gunshot residues (Line 101): “any surface in its vicinity”- be more specific here: on the hands of the shooter, clothes of the shooter, clothes of the victim, nearby surfaces, etc.
Instead of “These residues escape through the firearm's openings after discharge and can deposit on themselves or onto any surfaces in tis vicinity” the sentence was changed to “These residues escape through the firearm's openings after discharge and can deposit on the hands and cloths of the shooter, cloths of the victim or nearby surfaces” (line 103-105).
Inorganic Compounds (Line 112): “The inorganic compounds found in gunshot residues (IGSR)”- same issue with plurality
The word “residues” was replaced by “residue” (Line 116).
Inorganic Compounds (Lines 113-114): “lead styphnate and dioxide”- change it to “lead styphnate and lead dioxide”
The terms “lead styphnate and dioxide” were changed to “lead styphnate and lead dioxide” (Lines 117-118).
Inorganic Compounds (Lines 114-115): “Due to their high molecular mass and low abundance in nature, Sb, Ba and Pb…”- these compounds should be identified as antimony (Sb), barium (Ba), and lead (Pb) when mentioned for the first time.
The phrase “Due to their high molecular mass and low abundance in nature, Sb, Ba and Pb have been the main target of IGSR analysis”, now reads “Due to their high molecular mass and low abundance in nature, antimony (Sb), barium (Ba) and lead (Pb) have been the main target of IGSR analysis” (Lines 118-120).
Inorganic Compounds (Lines 135-136): “In these formulations heavy elements such as Pb, Sb, and Ba”- they changed the order of the common heavy metals; they should keep the order consistent with before: Sb, Ba and Pb
The sentence “In these formulations heavy elements such as Pb, Sb, and Ba”, was changed to "In these formulations heavy elements such as Sb, Ba, and Pb”, maintaining the order of the elements for consistency (Lines 140-141). This change in the order of the elements was also made in line number 192.
Potential sources of SG and GSR compounds (Line 182): “vehicle brakes linings [63,64]”- Raman spectroscopy has been reported (doi.org/10.1007/S00216-018-1359-1) does NOT show false positives due to vehicle brakes linings.
The sentence was removed from the manuscript since it does not add significant information to the document.
Potential sources of SG and GSR compounds (Line 189): “In addition, Pb it is still used in”- remove the “it”
The sentence “In addition, Pb it is still used in...” now reads “In addition, Pb is still used in...” (Line 194). This change of removing the word “it” was also made in line number 193.
Potential sources of SG and GSR compounds (Lines 198-202) “However, due to the removal of the signature-elements from the primer, and subsequently from the GSR particles and because the compounds used as replacement are common in the environment, the selection of new target analytes is a real necessity. Imposing also the adoption of with different analytical methodologies suitable for the analysis of these compounds [15,16]”- this is too long and wordy; also “the adoption of with different…”, the “with” needs to be removed
The paragraph “However, due to the removal of the signature-elements from the primer, and subsequently from the GSR particles and because the compounds used as replacement are common in the environment, the selection of new target analytes is a real necessity. Imposing also the adoption of with different analytical methodologies suitable for the analysis of these compounds [15,16]”, was modified and now reads “However, due to the removal of the signature-elements from the primer, there is a need for the selection of new target analytes. Because the compounds now used as replacement are relatively common in the environment, it is necessary to adopt different analytical methodologies suitable for the determination of these compounds [15,16].” (Lines 204-209)
Analysis of gunpowder and gunshot residues (Line 206): “Chemical analysis has been used to identify GSR particles in suspects…”- the word “in” is an odd word to use, I think “on” would be more appropriate; it is known that GSR can penetrate into some clothing, but for the most part, “on” would be most appropriate
In the phrase “Chemical analysis has been used to identify GSR particles in suspects…”, “in” was replaced by “on” and it now reads “Chemical analysis has been used to identify GSR particles on suspects…” (Line 214)
Analysis of gunpowder and gunshot residues (Lines 207-208): “However, over the last decade, research has been focusing on the expanding the applications of the chemical analysis”- in this sentence, remove the “the” in “the applications”- it has two redundant “the” in the sentence
In the sentence “However, over the last decade, research has been focusing on the expanding the applications of the chemical analysis...”, the two redundant “the” were removed, and it now reads “However, over the last decade, research has been focusing on expanding applications of the chemical analysis...” (Lines 217-218)
Analysis of gunpowder and gunshot residues (Lines 228-229): “they can only provide indicative results”- I don’t understand the use of “indicative” here. Maybe, “preliminary” is a better word here.
In the sentence “However, since most of these tests were designed specifically for field use, they can only provide indicative results, which considerably decreases their applicability to forensic ballistics [9,15,16].”, the term “indicative was replaced by “preliminary”, and it now reads “However, since most of these tests were designed specifically for field use, they can only provide preliminary results, which considerably decreases their applicability to forensic ballistics [9,15,16].” (Lines 237-240).
Analysis of gunpowder and gunshot residues (Line 297): “methyl-ethyl-ketone [22]”- consider this work here (doi.org/10.3390/chemosensors11010011).
The reference was added to the work, as [23] (Line 308).
Analysis of gunpowder and gunshot residues (Lines 335-336): “The authors focused on several OGSR using SPME”- I would add the word compounds after OGSR.
In the sentence “The authors focused on several OGSR using SPME combined with GC-TEA (thermal energy analysis) and GC-FID (flame ionisation detector) [82,103].” the word “compounds” was added after “OGSR”, and it now reads “The authors focused on several OGSR compounds using SPME combined with GC-TEA (thermal energy analysis) and GC-FID (flame ionisation detector) [82,103].” (Lines 347-349)
Analysis of gunpowder and gunshot residues (Line 386): “to thermally desorbed the extracted compounds for GC analyses…”- “desorbed” should not be past tense, change to “desorb”.
The sentence “However, to thermally desorbed the extracted compounds for GC analyses, a dedicated unit must be installed in the apparatus, seriously limiting the applicability of the methodology.” now reads “However, to thermally desorb the extracted compounds for GC analysis, a dedicated unit must be installed in the apparatus, seriously limiting the applicability of the methodology.” (Lines 421-424).
Analysis of gunpowder and gunshot residues (Lines 414-417): “Both methodologies can detect and identify IGSR and OGSR, even without heavy metals, in a fast, cost-effective, and non-destructive manner, producing IR spectra based on the radiation interaction with matter at a given wavenumber [72,114,118,119]”- the authors referenced both IR and Raman works, but then only said IR spectra; I would just remove “IR”.
The term “IR” was removed (Line 461).
Analysis of gunpowder and gunshot residues (Line 445): “They showed that the obtained spectrums had high similarity”- not “spectrums;” replace with “spectra”
The term “spectra” replaced “spectrums” (Line 491).
Analysis of gunpowder and gunshot residues (Line 461): “followed by a Raman spectroscopy to detect OGSR in the samples”- remove the “a” in the sentence
The word “a” was removed, and now the sentence reads “The procedure included a manual microscopical observation of GSR particles, followed by a Raman spectroscopy to detect OGSR in the samples …” (Lines 509-510)
Table 1, Raman and FTIR, Objectives “Comparison of IR profiles obtained from both…”- remove “IR”.
The term “IR” was removed (Line 552).
Analysis of gunpowder and gunshot residues (Lines 482-483): “The signal produced by these detectors is directly proportional to the compound's concentration on the sample”- consider changing “on” to “of”
The paragraph was rewritten, and the phrase was removed (Lines 556-559).
Analysis of gunpowder and gunshot residues (Lines 484-487): “Chromatography varies based on its stationary and mobile phases, being the most common in forensic laboratories those with a solid stationary phase and either a liquid mobile phase (liquid chromatography) or a gas mobile phase (gas chromatography)”- this a long run-on sentence that is oddly worded. How about… “Chromatography varies based on its stationary and mobile phases. In forensic science, there is a solid stationary phase with either a liquid mobile phase (liquid chromatography) or a gas mobile phase (gas chromatography)”
The paragraph was rewritten, and the phrase was removed (Lines 556-559).
Analysis of gunpowder and gunshot residues (Lines 535-536): “Gas Chromatography (GC) is among the most used techniques in the forensic context, in particular when coupled with mass spectroscopy (MS) [47,91,98,127].”- sounds a little odd; How about… “Gas Chromatography (GC) is among the most used technique in forensic science, especially when coupled with mass spectroscopy (MS)”
The sentence “Gas Chromatography (GC) is among the most used techniques in the forensic context, in particular when coupled with mass spectroscopy (MS) [47,91,98,127].” was altered, and now reads “Gas Chromatography (GC) is among the most used techniques in the forensic science, especially when coupled with mass spectroscopy (MS) [51,93,100,130].” (Line 626-628).
Analysis of gunpowder and gunshot residues (Lines 560-561): “This approach detected 32 OSGR in 9 mm cartridges after discharge over 32 hours.”- there should be a word following OGSR; I assume the authors mean OGSR compounds
The term “compounds” was added, and the sentence now reads “This approach detected 32 OSGR compounds in 9 mm cartridges after discharge over 32 hours.” (Line 652-653).
Analysis of gunpowder and gunshot residues (Lines 570-572): “This approach consisted of a capillary microextraction of volatiles (CMV) followed by GC-MS and laser-induced breakdown spectroscopy (LIBS) 571[130]” – It is important here to add the discussion of Raman and LIBS combination for the detection and analysis of GSR (doi.org/10.1016/j.saa.2023.122316).
A paragraph was added to address this comment, and it reads “More recently, Khandasammy et al. [125] applied Raman spectroscopy together with Laser-Induced Breakdown Spectroscopy (LIBS) to successful differentiate between the OGSR samples from ammunition types of the same caliber and produced by the same manufacturer. LIBS allows rapid, non-destructive, and simultaneous detection of multiple elements present in GSR samples. It requires minimal sample preparation, reducing analysis time and cost. This work showed that LIBS can aid in the identification and characterization of GSR components accurately. On the other hand, the laser-based nature of LIBS enables remote analysis, making it a promising technique suitable for field investigations.” (Lines 523-532).
Summary (Lines 689-690): “Allied to this, a standardized method for data analysis must also be defined” – the word “allied” is an odd choice; I would remove it entirely and say, “A standardized method for data analysis must also be defined”
The sentence was changed and now reads “In addition, a standardized method for data analysis must also be defined” (Lines 822).
Summary (Lines 694-695): “These alternative analytical methods can be extremely helpful in the forensic ballistics”- remove “the”
The term “the” was removed and the sentence now reads “These alternative analytical methods can be extremely helpful in forensic ballistics, particularly for firearm-related elements found in crime scenes where the conventional forensic ballistic visual comparison cannot be used.” (Lines 827-829).
Summary (Lines 698-703): “These include the development of predictive models to determine time since shooting and firing distances and to identifying ammunition’s manufacturer and its model, all taking advantage of the most recent statistical and machine learning tools currently available for model development and of modern computational database systems for the efficient storage and access to this data.”- this is one LONG sentence and has some grammar issues. How about… “These include the development of predictive models to determine time since shooting, firing distances, and to identify firearm manufacturer and its caliber. These methods can take advantage of the most recent statistical and machine learning tools currently available for model development and of modern computational database systems for the efficient storage and access to this data.”
The sentences were changed and now reads “These include the development of predictive models to determine time since shooting and firing distances, and to identify ammunition’s manufacturer and its model. These methods can take advantage of the most recent statistical and machine learning tools currently available for model development and of modern computational database systems for the efficient storage and data access.” (Lines 831-836).
Reviewer 2 Report
The structure of the paper is clear and well organized, with each section providing an in-depth discussion on its topic. This logical progression aids the reader in understanding the complexities of gunshot residue (GSR) analysis, from its composition to the current techniques and emerging applications in its analysis.
The manuscript thoroughly discusses the constituents of gunpowder and gunshot residues, giving particular emphasis on both inorganic and organic compounds. Given the recent legislative changes in some countries aiming to eliminate heavy metals from immunity, the authors h ave rightly identified the increasing importance of the analysis of organic gunshot residues (OGSR). Their comprehensive exploration of the existing and emerging analytical methods for OGSR is thus timely and highly relevant.
The authors' extensive discussion on the techniques used for the morphological and chemical analysis of GSR, including their detailed elaboration on solvent extraction, solid phase microextraction, headspace sorbtive extraction, spectroscopy, and chromatography, is part typically praiseworthy. The authors have skillfully identified the strengths and limitations of each method, a factor that will significantly contribute to the field.
The review section that explores emerging applications of GSR analysis techniques is very insightful. It provides a promising outlook on the future of forensic ballistics, highlighting the importance of methods like chromatography and spectroscopy for the analysis of OGSR.
In the conclusion, the authors eloquently summarize the importance of a paradigm shift in gunshot residue analysis, towards a greater emphasis on OGSR analysis and the need for the standardization of these methods. They highlight the potential benefits of combining IGSR and OGSR techniques and their potential applications in real-world scenarios.
My only suggestion would be to delve more into the comparative efficiency of combined IGSR and OGSR analysis, as this could potentially provide further insights into the superior methodology for gunshot residue analysis. Minor language errors need to be corrected.
A small number of errors do not affect the reader's understanding. They are not listed here. The use of some gerunds is problematic. Several usages of "analysis", and "analyze" are wrong.
Author Response
Response to reviewer #2 comments and suggestions
Overall comments and suggestions:
The structure of the paper is clear and well organized, with each section providing an in-depth discussion on its topic. This logical progression aids the reader in understanding the complexities of gunshot residue (GSR) analysis, from its composition to the current techniques and emerging applications in its analysis.
The manuscript thoroughly discusses the constituents of gunpowder and gunshot residues, giving particular emphasis on both inorganic and organic compounds. Given the recent legislative changes in some countries aiming to eliminate heavy metals from immunity, the authors have rightly identified the increasing importance of the analysis of organic gunshot residues (OGSR). Their comprehensive exploration of the existing and emerging analytical methods for OGSR is thus timely and highly relevant.
The authors' extensive discussion on the techniques used for the morphological and chemical analysis of GSR, including their detailed elaboration on solvent extraction, solid phase microextraction, headspace sorbtive extraction, spectroscopy, and chromatography, is part typically praiseworthy. The authors have skillfully identified the strengths and limitations of each method, a factor that will significantly contribute to the field.
The review section that explores emerging applications of GSR analysis techniques is very insightful. It provides a promising outlook on the future of forensic ballistics, highlighting the importance of methods like chromatography and spectroscopy for the analysis of OGSR.
In the conclusion, the authors eloquently summarize the importance of a paradigm shift in gunshot residue analysis, towards a greater emphasis on OGSR analysis and the need for the standardization of these methods. They highlight the potential benefits of combining IGSR and OGSR techniques and their potential applications in real-world scenarios.
My only suggestion would be to delve more into the comparative efficiency of combined IGSR and OGSR analysis, as this could potentially provide further insights into the superior methodology for gunshot residue analysis. Minor language errors need to be corrected.
Comments on the Quality of English Language
A small number of errors do not affect the reader's understanding. They are not listed here. The use of some gerunds is problematic. Several usages of "analysis", and "analyze" are wrong.
We appreciate the valuable feedback provided by the reviewer, which has significantly contributed to enhancing the quality of our manuscript. In response to the comments, we have included a new paragraph in the discussion section that explores the potential benefits and avenues for future research regarding the combined analysis of IGSR and OGSR compounds.
We have addressed the suggestion raised by the reviewer and added a new paragraph to the discussion section. This paragraph delves into the potential advantages of combining IGSR and OGSR analysis, opening up new avenues for research and practical applications in forensic laboratories. Various studies have demonstrated the successful utilization of alternative methods to detect, identify, and quantify OGSR compounds. However, researchers have noted that the efficiency can be improved by combining the analysis of both OGSR compounds and IGSR particles. Future research in this area could involve the integration of the standard IGSR particle analysis method, such as SEM-EDX, with chromatographic or spectroscopy techniques for the analysis of OGSR compounds. Techniques like Raman or FTIR spectroscopy, capable of analyzing both organic and inorganic GSR, have been suggested as potential tools for achieving this goal (references: 50, 74, 116, 120, 121). The combined analysis of organic and inorganic residues holds the promise of correlating samples with reference materials, thereby assisting in the investigation of crimes involving firearms (Lines 803-815).
Furthermore, we have thoroughly reviewed the manuscript and made necessary changes to the English language usage, specifically focusing on the consistent usage of the terms "analysis" and "analyses" throughout the paper
Reviewer 3 Report
.Reviewer comments
After making major changes in the article, I suggest submitting it to a forensic journal where it will have a wider audience of interested reader. My suggestion is supported by the 13 referred reviews on GSR, where 83% were published in forensic journals and the remaining 17% in specialized journals (for example, reference 76 is a review on electrochemical detection of GSR and is published in the journal Electroanalysis).
However, I should admit that I do not regard the present submission as a review article, but a primer educational article that aims at explaining the current techniques for GSR analysis to a less expert audience. For example, the chromatography section (subheading 3.2.1) reads:
“Chromatography is a robust chemical analysis technique that enables the separation of compounds from mixtures based on their physicochemical properties. In this technique, compounds are forced to travel through a specific matrix whose characteristics lead to different interactions with each of the individual compounds found in the mixture”.
or
“Liquid Chromatography (LC) is a highly sensitive and reproducible technique commonly used in forensic sciences. It is suited for separating non-volatile, semivolatile and thermolabile compounds and can be coupled with a wide range of detectors, conferring high flexibility to the technique”
In my opinion, the above information seems to be quoted from a basic guide for undergraduate students. This is not the kind of information that an average reader of Molecules will expect from a so-called review article, where the authors should analyse, discuss and compare in a very critical way the literature on methods and conclusions from previously published GSR studies. Unfortunately, the main goal of the submission is to give an account (either in text or table form) of what the researchers have done by means of different instrumental techniques, without any further insights or critical discussion.
The lack of critical inputs is so evident that for example the authors stated that one of the limitations of liquid chromatography (LC) is “… inability to sample gas samples, and its destructive nature”. After working for over 30 years in the subject it is the first time that I read that the inability of sampling gas samples is a disadvantage of LC. Are the authors aware that the technique is called “Liquid Chromatography” because is intended for liquids and not for gases? Are the authors aware that for gas samples they can use “Gas Chromatography”. In addition, are the authors aware that if they pass a compound through a LC column and detect its UV spectrum, they can collect the original compound after it passes the detector. Even though, if they use a mass detector, only one fraction of the sample is sent to mass analysis while a great percentage of the “intact sample” can be collected in an appropriate sample collector. Anyway, it is evident that the authors make comments without any further critical thinking.
The chemometrics approaches are poorly written. There are not specific insights, comprehensive comparisons, critical discussion about the effectivity of these techniques in GSR analysis. For instance, what is the impact of the number of GSR scores and loadings on the PCA discrimination process? This aspect will influence the reliability of the results and conclusions. Unfortunately, the authors only give a description of what every article is about without further ado. In addition, some statistical tools are mentioned without explaining their statistical power, or whether they are parametric or non-parametric, etc.
This manuscript does not meet the standards of Molecules, therefore I cannot recommend it for publication in its present form.
Specific comments
I suggest providing a list of abbreviations.
L20: State the full definition instead of the abbreviation “SEM-EDX”
L24: “… analysis of organic compound is leading to…” instead of “… analysis of organic compound is growing leading to…”
L61: “The present article will explore…” instead of “We will explore…”
L62: “this data”? data is the plural and datum the singular form, consequently you should state either “these data” or “this datum”
L64-66. Delete the paragraph “However, we will not explore sample collection procedures in depth. For a detailed review on the topic, please refer to Shrivastava et al. [14] or Goudsmits et al. [15].” After all you are not discussing any previous method or results in depth.
L68: “… is single-use cartridge” instead of “… is single-use cartridges”
L94-96: I suggest rephrasing as “These compounds significantly change the chemical and physical properties of gunpowder and its perform according to the specific purpose for which it was designed [15,24–27].” Instead of “These compounds significantly change the chemical and physical properties of the gunpowder and allow it to perform according to the specific purpose for which it was designed [15,24–27].”
L301-303: Considering the lack of an associated reference the following information is irrelevant and should be deleted: “Our research group also used solvent extraction in previous preliminary works, which included dichloromethane and methyl-ethyl-ketone to dissolve SG prior to GC-MS, Near-Infrared Spectroscopy (NIRS) and Fourier-Transform Infrared Spectroscopy (FTIR)”
L348 or L353: “…, they only noted…” or “…, they also…” avoid using personal pronouns.
L371: “80º C” underlined grade?
L417 or L445“…, they follow…” or “They showed…” avoid using personal pronouns.
L496: “…solvent, inability to sample…” instead of “…solvent, the inability to sample…”
L503: The authors state eight compound but I count only seven “(NG, DPA, AK II EC, 4-nDPA, 2-nDPA, 2,4-DNT, and N-n-DPA)”
L509-510: “Feeney et al. [36] developed and validated…” instead of “Also worth mentioning is the work of Feeney et al. [36], where they developed and validated…”
L517: the term LOD has not been defined before this line in the article.
L517: “… identify 32 OGSR compounds with lower LOD than previously reported” What is the significance of this statement? Well, neither this nor previous statements are critically discussed in this submission.
L531: Table 1 reference [50] reads in the conclusion column “…lower LOD thanks to ANN optimisation” It is the first time that I read an article that gives thanks to ANN which by the way has not been defined yet in the submission.
L579-593: This paragraph contains too many grammatical ornaments that makes the full information boring to read. I suggest going to the point in a straightforward manner.
Line 604: the reference order is incorrect. It should be: “[13,50,132,137–143,144,145,]” instead of
“[13,50,144,145,132,137–143]”
L674: The summary should be shortened and discussed appropriately, I only read what the researchers have done without any further critical comments. It is a very boring section.
L670: There is a Table 4 that gives further account of what the researchers have done, however the table in question is not referred in the text.

The article is difficult to read due to the recurrent use of excessively ornamented sentences. For instance, the abstract section reads:
“The identification of firearms is of paramount importance for investigating crimes involving firearms, as it establishes the link between a particular firearm and firearm-related elements found at a crime scene…”
This quite repetitive sentence can be summarised as:
“The identification of firearms used in crimes and their connection with crime scene evidence is of importance in forensic science”.
Scientific English style should be concise and avoid using florid sentences and personal pronouns (e.g., we, they). These issues are found throughout the submission.
Author Response
Response to reviewer #3 comments and suggestions
Overall comments
After making major changes in the article, I suggest submitting it to a forensic journal where it will have a wider audience of interested reader. My suggestion is supported by the 13 referred reviews on GSR, where 83% were published in forensic journals and the remaining 17% in specialized journals (for example, reference 76 is a review on electrochemical detection of GSR and is published in the journal Electroanalysis).
However, I should admit that I do not regard the present submission as a review article, but a primer educational article that aims at explaining the current techniques for GSR analysis to a less expert audience. For example, the chromatography section (subheading 3.2.1) reads:
“Chromatography is a robust chemical analysis technique that enables the separation of compounds from mixtures based on their physicochemical properties. In this technique, compounds are forced to travel through a specific matrix whose characteristics lead to different interactions with each of the individual compounds found in the mixture”.
or
“Liquid Chromatography (LC) is a highly sensitive and reproducible technique commonly used in forensic sciences. It is suited for separating non-volatile, semivolatile and thermolabile compounds and can be coupled with a wide range of detectors, conferring high flexibility to the technique”
In my opinion, the above information seems to be quoted from a basic guide for undergraduate students. This is not the kind of information that an average reader of Molecules will expect from a so-called review article, where the authors should analyse, discuss and compare in a very critical way the literature on methods and conclusions from previously published GSR studies. Unfortunately, the main goal of the submission is to give an account (either in text or table form) of what the researchers have done by means of different instrumental techniques, without any further insights or critical discussion.
The lack of critical inputs is so evident that for example the authors stated that one of the limitations of liquid chromatography (LC) is “… inability to sample gas samples, and its destructive nature”. After working for over 30 years in the subject it is the first time that I read that the inability of sampling gas samples is a disadvantage of LC. Are the authors aware that the technique is called “Liquid Chromatography” because is intended for liquids and not for gases? Are the authors aware that for gas samples they can use “Gas Chromatography”. In addition, are the authors aware that if they pass a compound through a LC column and detect its UV spectrum, they can collect the original compound after it passes the detector. Even though, if they use a mass detector, only one fraction of the sample is sent to mass analysis while a great percentage of the “intact sample” can be collected in an appropriate sample collector. Anyway, it is evident that the authors make comments without any further critical thinking.
The chemometrics approaches are poorly written. There are not specific insights, comprehensive comparisons, critical discussion about the effectivity of these techniques in GSR analysis. For instance, what is the impact of the number of GSR scores and loadings on the PCA discrimination process? This aspect will influence the reliability of the results and conclusions. Unfortunately, the authors only give a description of what every article is about without further ado. In addition, some statistical tools are mentioned without explaining their statistical power, or whether they are parametric or non-parametric, etc.
This manuscript does not meet the standards of Molecules, therefore I cannot recommend it for publication in its present form.
We sincerely appreciate your comments and suggestions, which have been invaluable in improving the manuscript. We would like to address your concerns and provide clarifications on the points you raised.
Regarding your suggestion to submit the article to a forensic journal, we would like to mention that our intention was to contribute to the special issue of Molecules focusing on forensic analysis in chemistry. While we acknowledge the wider audience of forensic journals, we believe that this special issue can attract readers from the forensic sciences who may not be as familiar with these methodologies as the typical readers of Molecules. Therefore, we feel that the manuscript aligns well with the scope and objective of the special issue.
We have made significant changes throughout the manuscript, including addressing the English language issues highlighted by the reviewer.
We have taken your feedback into account and provided more discussion on selected topics, offering further insights and critical analysis. Check lines 391-408, 440-444, 516-550, 616-619, 663-669, 727-732, among many others.
Regarding the statements about chromatographic techniques, particularly liquid chromatography (LC), we acknowledge that they may appear basic to someone well-versed in the subject. However, our intention was to provide clear explanations for readers who may not be familiar with these techniques, particularly those who are less experienced or new to the field. We understand that the statements might be simplified, and we have made revisions to strike a balance between accessibility and scientific accuracy.
In response to your comments on the chemometric approaches, we have expanded the discussion to provide more insights and critical analysis of their effectiveness in GSR analysis.
We believe that the revised manuscript meets the standards of Molecules, especially considering the focus of the special issue on forensic analysis in chemistry. The manuscript presents a comprehensive review of a hot topic in chemistry applied to the forensic field, and we are confident that it can contribute significantly to the scientific community.
To address the specific concerns and comments raised by the reviewer in a comprehensive manner, we have provided a detailed point-by-point response below.
Specific comments
I suggest providing a list of abbreviations.
An abbreviations list was not provided since it is not costumary in Molecules.
L20: State the full definition instead of the abbreviation “SEM-EDX”
Full definition was provided. Lines 20-21.
L24: “… analysis of organic compound is leading to…” instead of “… analysis of organic compound is growing leading to…”
The sentence was revised. Lines 24-25.
L61: “The present article will explore…” instead of “We will explore…”
The sentence was revised. Lines 63-64.
L62: “this data”? data is the plural and datum the singular form, consequently you should state either “these data” or “this datum”
The sentence was revised. Line 65.
L64-66. Delete the paragraph “However, we will not explore sample collection procedures in depth. For a detailed review on the topic, please refer to Shrivastava et al. [14] or Goudsmits et al. [15].” After all you are not discussing any previous method or results in depth.
The sentence was not removed, since it provides the reader with additional references if they wish to explore the topic further. Lines 67-69.
L68: “… is single-use cartridge” instead of “… is single-use cartridges”
The sentence was revised. Line 71.
L94-96: I suggest rephrasing as “These compounds significantly change the chemical and physical properties of gunpowder and its perform according to the specific purpose for which it was designed [15,24–27].” Instead of “These compounds significantly change the chemical and physical properties of the gunpowder and allow it to perform according to the specific purpose for which it was designed [15,24–27].”
The sentence was revised. Line 97-99.
L301-303: Considering the lack of an associated reference the following information is irrelevant and should be deleted: “Our research group also used solvent extraction in previous preliminary works, which included dichloromethane and methyl-ethyl-ketone to dissolve SG prior to GC-MS, Near-Infrared Spectroscopy (NIRS) and Fourier-Transform Infrared Spectroscopy (FTIR)”
We agree and removed the sentence.
L348 or L353: “…, they only noted…” or “…, they also…” avoid using personal pronouns.
The sentences were revised.
L371: “80º C” underlined grade?
The sentence was revised. Line 386.
L417 or L445“…, they follow…” or “They showed…” avoid using personal pronouns.
The sentences were revised.
L496: “…solvent, inability to sample…” instead of “…solvent, the inability to sample…”
The sentence was revised. Line 579.
L503: The authors state eight compound but I count only seven “(NG, DPA, AK II EC, 4-nDPA, 2-nDPA, 2,4-DNT, and N-n-DPA)”
The sentence was revised. Lines 586-587.
L509-510: “Feeney et al. [36] developed and validated…” instead of “Also worth mentioning is the work of Feeney et al. [36], where they developed and validated…”
The sentence was revised. Lines 593-594.
L517: the term LOD has not been defined before this line in the article.
The sentence was revised. Line 601.
L517: “… identify 32 OGSR compounds with lower LOD than previously reported” What is the significance of this statement? Well, neither this nor previous statements are critically discussed in this submission.
The sentence was revised. Lines 601-603.
L531: Table 1 reference [50] reads in the conclusion column “…lower LOD thanks to ANN optimisation” It is the first time that I read an article that gives thanks to ANN which by the way has not been defined yet in the submission.
The abbreviation is now defined. The information is accurate. Check DOI: 10.1039/c5ay00306g. Table 2
L579-593: This paragraph contains too many grammatical ornaments that makes the full information boring to read. I suggest going to the point in a straightforward manner.
The paragraph was revised. Lines 676-690.
Line 604: the reference order is incorrect. It should be: “[13,50,132,137–143,144,145,]” instead of “[13,50,144,145,132,137–143]”
The reference order was corrected. was revised. Line 702.
L674: The summary should be shortened and discussed appropriately, I only read what the researchers have done without any further critical comments. It is a very boring section.
The summary section was extensively revised. Lines 809-852.
L670: There is a Table 4 that gives further account of what the researchers have done, however the table in question is not referred in the text.
Table 4 is now referenced in the text. Line 688.
Comments on the Quality of English Language
The article is difficult to read due to the recurrent use of excessively ornamented sentences. For instance, the abstract section reads:
“The identification of firearms is of paramount importance for investigating crimes involving firearms, as it establishes the link between a particular firearm and firearm-related elements found at a crime scene…”
This quite repetitive sentence can be summarised as:
“The identification of firearms used in crimes and their connection with crime scene evidence is of importance in forensic science”.
Scientific English style should be concise and avoid using florid sentences and personal pronouns (e.g., we, they). These issues are found throughout the submission.
The manuscript was extensively revised regarding English language and editing.